# Prodynorphin and Proenkephalin in Cerebrospinal Fluid of Sporadic Creutzfeldt–Jakob Disease

**DOI:** 10.3390/ijms23042051

**Published:** 2022-02-12

**Authors:** Samir Abu-Rumeileh, Peggy Barschke, Patrick Oeckl, Simone Baiardi, Angela Mammana, Andrea Mastrangelo, Mhd Rami Al Shweiki, Petra Steinacker, Anna Ladogana, Sabina Capellari, Markus Otto, Piero Parchi

**Affiliations:** 1Department of Neurology, Martin-Luther-University Halle-Wittenberg, 06120 Halle (Saale), Germany; samir.aburumeileh@uk-halle.de (S.A.-R.); petra.steinacker@uk-halle.de (P.S.); 2Department of Neurology, Ulm University Hospital, 89081 Ulm, Germany; barschke.peggy@gmx.de (P.B.); patrick.oeckl@uni-ulm.de (P.O.); ramishweiki@gmail.com (M.R.A.S.); 3German Center for Neurodegenerative Diseases (DZNE e.V.), 89081 Ulm, Germany; 4IRCCS Istituto delle Scienze Neurologiche di Bologna, 40139 Bologna, Italy; simone.baiardi6@unibo.it (S.B.); angela.mammana@studio.unibo.it (A.M.); andrea.mastrangelo4@studio.unibo.it (A.M.); sabina.capellari@unibo.it (S.C.); 5Department of Experimental Diagnostic and Specialty Medicine (DIMES), University of Bologna, 40139 Bologna, Italy; 6Department of Biomedical and NeuroMotor Sciences (DIBINEM), University of Bologna, 40139 Bologna, Italy; 7Department of Neuroscience, Istituto Superiore di Sanità, 00161 Rome, Italy; anna.ladogana@iss.it

**Keywords:** biomarkers, prion disease, opioid peptides, mass spectrometry, proenkephalin, prodynorphin, Creutzfeldt–Jakob disease

## Abstract

Proenkephalin (PENK) and prodynorphin (PDYN) are endogenous opioid peptides mainly produced in the striatum and, to a lesser extent, in the cerebral cortex. Dysregulated metabolism and altered cerebrospinal fluid (CSF) levels of PENK and PDYN have been described in several neurodegenerative diseases. However, no study to date investigated these peptides in the CSF of sporadic Creutzfeldt–Jakob disease (sCJD). Using liquid chromatography-multiple reaction monitoring mass spectrometry, we evaluated the CSF PDYN- and PENK-derived peptide levels in 25 controls and 63 patients with sCJD belonging to the most prevalent molecular subtypes (MM(V)1, VV2 and MV2K). One of the PENK-derived peptides was significantly decreased in each sCJD subtype compared to the controls without a difference among subtypes. Conversely, PDYN-derived peptides were selectively decreased in the CSF of sCJD MV2K, a subtype with a more widespread overall pathology compared to the sCJD MM(V)1 and the VV2 subtypes, which we confirmed by semiquantitative analysis of cortical and striatal neuronal loss and astrocytosis. In sCJD CSF PENK and PDYN were associated with CSF biomarkers of neurodegeneration but not with clinical variables and showed a poor diagnostic performance. CSF PDYN and PENK-derived peptides had no significant diagnostic and prognostic values in sCJD; however, the distinct marker levels between molecular subtypes might help to better understand the basis of phenotypic heterogeneity determined by divergent neuronal targeting.

## 1. Introduction

Sporadic Creutzfeldt–Jakob disease (sCJD) is the most common human prion disease and includes six distinct clinicopathological subtypes that are mainly determined by the genotype at the methionine (M)/valine (V) polymorphic codon 129 of the *PRNP* gene and type (1 or 2) of the disease-associated prion protein (PrP^Sc^) accumulating in the brain. They have been named accordingly as MM(V)1, MM2 with predominant cortical pathology (MM2C), MM2 with predominant thalamic degeneration (MM2T), MV2 with kuru-type amyloid plaques (MV2K), VV1 and VV2 [1,2].

Several biofluid markers of neuronal damage, neuroinflammation and synaptic dysfunction have been evaluated in sCJD, aiming to improve diagnosis, prognostic evaluation, stratification and management of patients [3,4,5,6]. However, the continuous identification of new potential cerebrospinal fluid (CSF) biomarkers is still mandatory to achieve a better understanding of other pathogenetic pathways involved in sCJD.

In recent years, the dysregulated metabolism of neuropeptides in neurodegenerative diseases (NDs) has attracted growing attention. Endogenous opioids are a group of peptides that act on opioids receptors and that derive from proteolytic cleavage of three main precursors: proenkephalin (PENK), prodynorphin (PDYN) and pro-opio-melanocortin [7]. Altered levels of CSF PDYN and/or PENK-derived peptides have been reported in Alzheimer’s disease (AD), frontotemporal dementia (FTD), dementia with Lewy bodies (DLB) and Huntington’s disease (HD) [8,9,10,11,12]. 

Initial evidence from these studies suggested that decreased CSF PDYN and PENK levels may reflect an impairment and/or neurodegeneration of the striatal medium spiny projection neurons (MSNs), which produce both peptides under dopaminergic signaling [9,10]. Moreover, dysfunctions in the PDYN pathway appear to be involved in developing behavioral and sleep disorders in neurodegenerative disease [12]. Nevertheless, no study to date has evaluated CSF PDYN and PENK in sCJD, a highly heterogeneous disease from both clinical and neuropathological points of view.

Using our previously developed liquid chromatography-tandem mass spectrometry (LC-MS/MS) method in multiple reaction monitoring (MRM) mode for the measurement of CSF PDYN-derived peptides [9] and a new assay for PENK-derived peptides, we investigated the profiles of these markers in sCJD, including its molecular subtypes, and studied the possible associations between the neuropeptide levels and those of other biomarkers and clinical variables, such as disease stage and survival.

## 2. Results

Sporadic CJD cases and controls showed no difference in sex, but sCJD cases were slightly (but not with statistically significance) older than the controls (Table 1). There was no effect of sex and age on CSF biomarker levels (Appendix A).

Sporadic CJD patients showed no difference in both PDYN-derived peptide levels compared to the controls, whereas the PENK peptide [DAE...LLK], but not [FAE...YSK], was significantly lower in sCJD compared with the controls (*p* < 0.001) (Table 1 and Figure 1). These findings were confirmed even after age-adjustment or using the mean of the two peptides for the calculations as described [9] (Appendix A). 

PDYN and PENK peptides showed a suboptimal diagnostic accuracy with AUC < 0.80 in the discrimination between sCJD and controls except for the PENK peptide [DAE...LLK], which yielded a borderline performance (0.768 ± 0.061) (Appendix A). After stratification according to the most prevalent sCJD subtypes, the MV2K group showed significantly lower PDYN [SVG...LAR] levels compared to the VV2 (*p* = 0.034) group and significantly lower PDYN [FLP...STR] levels than the VV2 (*p* = 0.021) and MM(V)1 (*p* = 0.011) groups (Table 2 and Figure 1). 

Similarly, both PDYN peptides were significantly lower in MV2K patients than in the controls ([SVG...LAR]: *p* = 0.019, [FLP...STR]: *p* = 0.048). PENK [DAE...LLK] and [FAE...YSK] peptide levels did not differ among the sCJD most prevalent subtypes. However, the PENK [DAE...LLK] peptide was significantly decreased in each sCJD subtype compared to the controls (MM(V)1: *p* < 0.001, VV2: *p* = 0.012, MV2K: *p* = 0.037) (Table 2 and Figure 1). All results were confirmed even after age adjustment and by including the mean of each couple of peptides in the calculations (Appendix A).

All peptides showed a suboptimal accuracy in the discrimination between sCJD subtypes and controls and among sCJD subtypes with an AUC < 0.80 except for a borderline performance of PENK peptide [DAE...LLK] in the comparison between sCJD MM(V)1 and controls (AUC 0.844 ± 0.077) (Appendix A).

In the semiquantitative neuropathological analysis, we observed more pronounced striatal astrogliosis and neuronal loss in sCJD MV2K (4.4 ± 0.9) and VV2 (4.4 ± 0.8) than in sCJD MM(V)1 (3.2 ± 0.9, *p* = 0.0005 and *p* = 0.0002, respectively). In contrast, sCJD MV2K (3.5 ± 1.7) and MM(V)1 (3.0 ± 0.8) showed more severe neuropathological changes in the cerebral cortex compared to VV2 cases (1.6 ± 0.9, *p* = 0.0001 and *p* = 0.0005, respectively) (Figure 2). Overall, there was a significantly higher amount of corticostriatal neuronal loss and gliosis in the sCJD MV2K subtype than in the other two subtypes (MV2K 7.8 ± 2.5 vs. VV2 6.0 ± 1.6, *p* = 0.0317; vs. MM(V)1 6.1 ± 1.7, *p* = 0.0202) (Figure 2).

In the sCJD group, we detected strong correlations between the levels of the two PDYN- (*r* = 0.614, *p* < 0.001) and the two PENK-derived peptides (*r* = 0.734, *p* < 0.001), suggesting a consistent and reliable estimate of both opioid peptide levels in the CSF. The same was also confirmed in the control group (PDYN: *r* = 0.661, *p* < 0.001, PENK: *r* = 0.673, *p* = 0.001).

The same was also confirmed in the control group (PDYN: *r* = 0.661, *p* < 0.001, PENK: *r* = 0.673, *p* = 0.001). Moreover, PDYN-derived peptides correlated with t-tau ([SVG...LAR]: *r* = 0.459, *p* < 0.001; [FLP...STR]: *r* = 0.419, *p* < 0.001), NfL ([SVG...LAR]: *r* = 0.439, *p* < 0.001) and protein 14-3-3 ([SVG...LAR]: *r* = 0.441, *p* < 0.001; [FLP...STR]: *r* = 0.413, *p* = 0.001) in the sCJD group. In the same group, the PENK [DAE...LLK] peptide showed an association with YKL-40 (*r* = 0.587, *p* = 0.005) (Appendix A). CSF median levels of total(t)-tau, neurofilament light chain protein (NfL), protein 14-3-3 and chitinase-3-like protein 1 (YKL-40) in the sCJD group are shown in Appendix A. There was no correlation between neuropeptide levels and disease stage.

Based on univariate Cox regression analyses (63 prion disease patients, 63 dead and 0 censored), we identified the age (HR (CI 95%) 1.073 (1.033–1.114), *p* < 0.001), time from onset to lumbal puncture (LP) (HR (CI 95%) 0.661 (0.551–0.793), *p* < 0.001) and molecular subtype (VV2 vs. MM(V)1, HR (CI 95%) 0.523 (0.277–0.969), *p* = 0.043; MV2K vs. MM(V)1 HR (CI 95%) 0.122 (0.051–0.294), *p* < 0.001) as predictors of the mortality hazard ratios in sCJD, whereas the concentration of each peptide was not associated with survival.

## 3. Discussion

In the present study, we investigated, for the first time, the levels of the PDYN- and PENK-derived peptides in patients with sCJD to thereby provide new insights into the possible mechanisms underlying the dysregulation of brain opioids in neurodegenerative diseases.

Brain opioid peptides play an important role in the striatal network, which consists of MSNs and GABAergic interneurons [13]. MSNs comprised two sub-populations, belonging to the direct and the indirect basal ganglia pathways expressing PDYN and PENK, respectively. However, PDYN is also expressed the cerebral cortex at levels almost comparable to those of the striatum [7,14].

In our study, we showed that both PDYN-derived peptides were more significantly decreased in sCJD MV2K, a subtype with a more severe combined cortico-striatal pathology compared to the typical sCJD MM(V)1 and the VV2 subtypes [2,15], which we confirmed in the present cohort using a semiquantitative neuropathological analysis. Thus, the neurodegeneration of striatal MSNs combined with the cortical neuronal loss may be responsible for the PDYN-derived peptide decrease in the CSF of MV2K. 

Accordingly, decreased CSF PDYN levels have been also reported in HD and DLB [9,12,16]—proteinopathies with a prominent striatal or cortico-striatal pathology [9,16]. Since sCJD MV2K and MM(V)1 showed comparable reduced levels of PDYN [SVG...LAR] compared to the VV2 groups, correlating with a higher neuropathological score in the cerebral cortex, it is plausible that the degree of cortical pathology significantly affects this peptide’s concentration in the CSF.

Concerning PENK, we found a decrease of the PENK [DAE...LLK] peptide in all sCJD subtypes compared to the controls without differences among subgroups. From the biochemical point of view, the two PENK-derived peptides include the amino acidic residues 142–157 and 236–251. Given that the protein precursor PENK contains several cleavage sites between the amino acid residues 157 and 236 [17], the measured peptides in the CSF belong to distinct cleavage products with possibly different pathophysiological destinies. 

However, apart from sCJD patients, decreased or a trend towards decreased CSF PENK levels were also previously reported in HD, AD, FTD and DLB, suggesting a common pathogenetic pathway in most proteinopathies [8,10]. Further study on larger patient cohorts will be needed to fully elucidate the precise role of brain opioid peptides in neurodegenerative diseases.

Our findings of positive correlations between the peptides and biomarker of neuronal damage supports the link between ongoing neurodegeneration and CSF PDYN and PENK alterations. Nevertheless, the lack of association between CSF PDYN and PENK levels and disease duration in sCJD appears to exclude an influence of the disease progression rate on the CSF levels of brain opioid peptides. However, the many variables, including the wide phenotypic heterogeneity influencing disease duration in sCJD, would require a much more in-depth analysis in a larger cohort to reach a more definite conclusion on this issue.

The major strength of our study is the inclusion of all most prevalent subtypes of sCJD with a large majority of cases being autopsy-verified. On the other side, the implementation of mass-spectrometry-based biomarkers in clinical routines is currently hampered by the limited availability of the technique.

Despite the overall limited diagnostic and prognostic roles of PDYN and PENK in sCJD, our study showed differentially altered levels of PENK- and PDYN-derived peptides in most prevalent sCJD subtypes. These findings might reflect distinct degrees of striatal and cortical pathology in the sCJD spectrum and might help to better understand the disease neuropathological heterogeneity for the development of targeted therapies.

## 4. Materials and Methods

### 4.1. Patient Selection, CSF and Neuropathological Analyses

For the present study, we selected CSF samples of 63 patients with sCJD and 25 healthy controls that had been submitted to the Neuropathology Laboratory (NP-Lab) at the Institute of Neurological Sciences of Bologna (Italy) or to the National CJD Surveillance Unit at the Istituto Superiore di Sanità in Rome (Italy) from 2009 to 2021 for diagnostic activity related to CJD surveillance. The study was conducted in accordance with the Declaration of Helsinki and approved by the Istituto Superiore di Sanità Ethics Committee (approval number CE-ISS 09/266; 29 May 2009). Informed consent was given by study participants or by their next of kin.

The diagnosis of sCJD fulfilled the current European diagnostic criteria [18] and neuropathological consensus criteria [19]. The sCJD group comprised 56 cases with a definite neuropathological diagnosis and seven cases with a probable diagnosis of sCJD (all tested positive by CSF prion real-time quaking-induced conversion (RT-QuIC) assay) [20,21]. Demographic and clinical data of the sCJD cohort are shown in Appendix A. For the analysis according to the sCJD molecular subtypes, we merged the subjects with definite sCJD (28 MM(V)1, 15 VV2 and 13 MV2K) with those with a probable sCJD diagnosis (five VV2 and two MV2K) and a high level of certainty for a given subtype as described [22]. In each case, we calculated the length of survival and the disease stage as described [23]. The control group included 25 subjects lacking any clinical or neuroradiologic evidence of central nervous system disease and having CSF p-tau, t-tau and Aβ42 in the normal range [22].

CSF samples were obtained by LP following a standard procedure, centrifuged and stored at −80 °C. CSF PDYN- and PENK-derived peptides were analyzed using LC-MS/MS at Experimental Neurology Laboratory at Ulm University Hospital in all cases as described [9]. A detailed description of the sample preparation and MRM method is provided in the Appendix A. CSF t-tau, protein 14-3-3, NfL and YKL-40 were analyzed in the sCJD group and AD core biomarkers in the control group at the NP-Lab of Bologna using commercially available ELISA kits as already published [22]. PrP^Sc^ seeding activity was detected by RT-QuIC [20,21].

Paraffin-embedded block sections from the frontal (middle and superior frontal gyri), temporal (middle and superior temporal gyri), parietal (inferior and superior parietal lobes) and occipital cortices (calcarine and associative) and striatum (caudate and putamen nuclei) of definite sCJD MM(V)1 (*n* = 28), VV2 (*n* = 15) and MV2K (*n* = 13) brains were stained with hematoxylin-eosin and assessed semiquantitatively for neuronal loss and astrogliosis at the NP-Lab [24]. 

We provided a mean combined score for each case based on two operators’ semi-quantitative assessments of neuronal loss and astrogliosis (0, no significant pathology; 1, mild; 2, moderate; and 3, severe, see Appendix A) in the striatum (i.e., a mean score of the caudate and putamen) and cerebral cortex (i.e., a mean score of the frontal, temporal, parietal and occipital lobes).

### 4.2. Statistical Analyses

Statistical analyses were performed using IBM SPSS Statistics version 21 (IBM, Armonk, NY, USA) and GraphPad Prism 7 (GraphPad Software, La Jolla, CA, USA) software. For continuous variables, depending on the data distribution and the number of groups, we applied the Mann–Whitney U test, *t*-test, Kruskal–Wallis test (followed by Dunn–Bonferroni post hoc test) or the one-way analysis of variance (followed by Tukey’s post hoc test). All reported *p*-values were adjusted for multiple comparisons. The Chi-Square test was adopted for categorical variables. 

We used multivariate linear regression models to adjust for age differences in CSF biomarkers between the groups and used Spearman’s correlations to test the possible associations between analyzed variables. We also performed univariate/multivariate Cox regression analysis to test the association between marker levels and survival as well as other well-known prognostic factors [23,25]. The results are presented as Hazard Ratios (HRs) and 95% confidence intervals (95% CIs). Differences were considered statistically significant at *p* < 0.05.

## Figures and Tables

**Figure 1 ijms-23-02051-f001:**
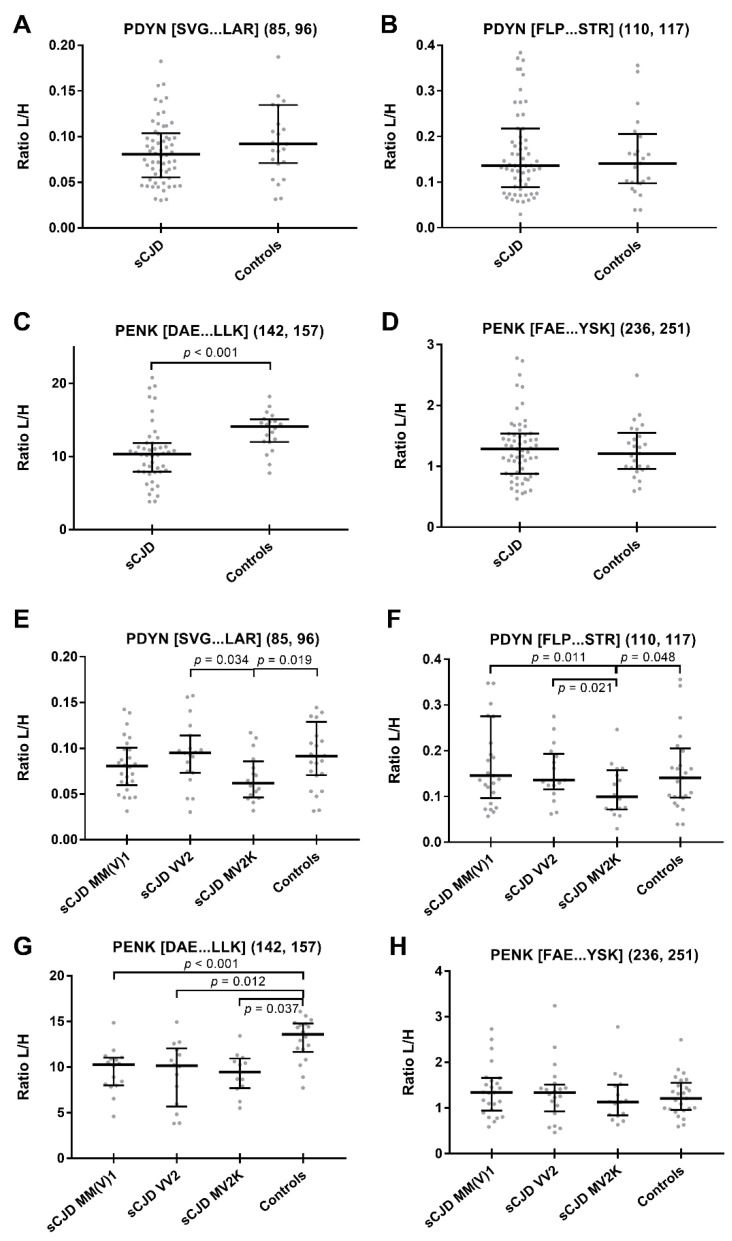
CSF levels of PENK and PDYN in sCJD, its subtypes and controls. Levels were determined by the measurement of two PDYN peptides ([SVG...LAR] and [FLP...STR]) and two PENK peptides ([DAE...LLK] and [FAE...YSK]) by targeted liquid chromatography-tandem mass spectrometry. The median and interquartile range are shown for the ratio of light peptides to spiked heavy labelled peptides (L/H). Kruskal–Wallis test and Dunn’s post hoc test. Comparisons of biomarker levels between sCJD and controls are shown in subfigures (**A**–**D**), whereas comparisons among sCJD subtypes and controls are shown in subfigures (**E**–**H**).

**Figure 2 ijms-23-02051-f002:**
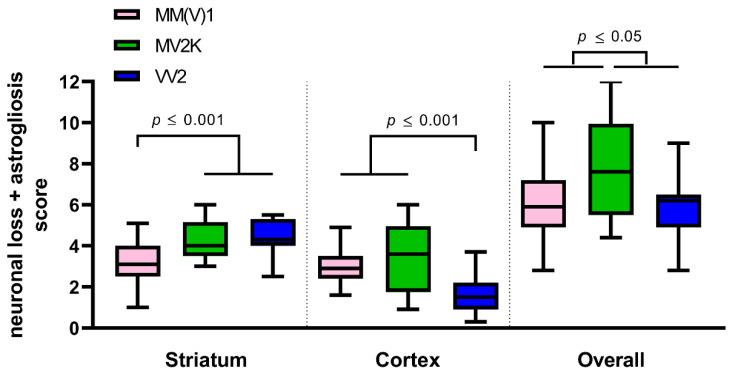
Semiquantitative neuropathological analysis of neuronal loss and astrogliosis in the striatum and cortex of sCJD subtypes MM(V)1, MV2K and VV2. Comparisons between sCJD subtypes MM(V)1, MV2K and VV2 concerning the score of neuronal loss and astrogliosis in the striatum and cerebral cortex. Kruskal–Wallis test and Dunn’s post hoc test.

**Table 1 ijms-23-02051-t001:** Demographic data and PDYN and PENK levels in the diagnostic groups.

Diagnosis	sCJD	Controls	*p*
N	63	25	
Age at LP (years ± SD)	70.22 ± 8.32	66.36 ± 7.59	0.066
Female (%)	46.0%	47.0%	0.642
Time from symptom onset to LP (months ± SD)	3.41 ± 2.69	-	-
PDYN [SVG…LAR] L/H ratio	0.082 (0.056–0.103)	0.092 (0.071–0.135)	0.126
PDYN [FLP…STR] L/H ratio	0.137 (0.090–0.218)	0.141 (0.098–0.206)	0.760
Mean PDYN L/H ratio	0.116 (0.072–0.172)	0.127 (0.080–0.183)	0.691
PENK [DAE…LLK] L/H ratio	10.266 (7.902–11.864)	14.341 (12.014–15.399)	<0.001
PENK [FAE…YSK] L/H ratio	1.287 (0.877–1.541)	1.211 (0.960–1.551)	0.835
Mean PENK L/H ratio	5.731 (4.391–6.632)	7.619 (6.612–8.319)	0.001

CSF levels of opioid peptides are given as the median and interquartile range. L/H: ratio of light peptides to spiked heavy labelled peptides. Abbreviations: sCJD, sporadic Creutzfeldt–Jakob disease; LP, lumbar puncture; PDYN, prodynorphin; PENK, proenkephalin; and SD, standard deviation.

**Table 2 ijms-23-02051-t002:** Distribution of PDYN- and PENK-derived peptides among sCJD subtypes.

Diagnosis	MM(V)1	VV2	MV2K
N	28	20	15
Age at LP (years ± SD)	70.39 ± 9.57	71.70 ± 7.90	67.93 ± 6.05
Female (%)	53.6	35.0	66.7
Time from symptom onset to LP (months ± SD)	1.95 ± 1.25	3.28 ± 1.13	6.19 ± 3.80
Survival (months ± SD)	3.25 ± 1.47	5.57 ± 2.08	17.46 ± 14.86
PDYN [SVG…LAR] L/H ratio	0.081 (0.059–0.104)	0.095 (0.074–0.122)	0.064 (0.046–0.088)
PDYN [FLP…STR] L/H ratio	0.149 (0.091–0.296)	0.149 (0.124–0.241)	0.104 (0.071–0.161)
Mean PDYN L/H ratio	0.119 (0.071–0.208)	0.126 (0.102–0.197)	0.092 (0.062–0.119)
PENK [DAE…LLK] L/H ratio	8.910 (7.915–11.064)	10.486 (6.437–12.705)	10.521 (8.397–14.130)
PENK [FAE…YSK] L/H ratio	1.337 (0.830–1.642)	1.336 (0.926–1.516)	1.138 (0.855–1.522)
Mean PENK L/H ratio	5.003 (4.374–6.2772)	5.772 (3.690–6.982)	5.852 (4.6808–8.070)

CSF levels of opioid peptides are given as the median and interquartile range. L/H: ratio of light peptides to spiked heavy labelled peptides. Abbreviations: LP, lumbar puncture; PDYN, prodynorphin; PENK, proenkephalin; and SD, standard deviation.

## Data Availability

The data presented in this study are available on request from the corresponding author.

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
