# Peer review of "Prodynorphin and Proenkephalin in Cerebrospinal Fluid of Sporadic Creutzfeldt–Jakob Disease"

_ijms, 2022, doi:10.3390/ijms23042051_

Round 1

Reviewer 1 Report

In this study, the authors by Abu-Rumeileh et al. have investigated the level of PENK- and PDYN-derived peptides between CJD patients and controls. The results suggest that one of the PENK-derived peptides was significantly decreased in each CJD subtype compared to controls.

The data are clearly presented and the interpretation is reasonable.

There are some major and minor points that the authors may consider:

Major comment

1) The authors analyzed the levels of PENK peptide and PDYN peptide separately from patients and controls (Tables 1,2 and Figure 1). What would be the result if the two levels were combined and analyzed?

2) The authors conducted a study in 66 CJD patients. Among them, the patients with sCJD and fCJD were 63 and 3, respectively. In my opinion, I recommend to analyze excluding three fCJDs, which account for a small part, and then change CJD to sCJD in the title and all contents.

Minor comments

Line 34,  did not associate -> were not associated.

Line 44, PRNP -> PRNP

Line 193, PRNP -> PRNP

Line 194,  In a sentence “3 carrying a pathogenic PRNP mutation (E200K, V210I, D178N, P102L)”, there are 4 mutation numbers. But why are 3 patients? Patient carrying a PRNP P102L mutation was GSS, not CJD.

Author Response

Reviewer 1.

In this study, the authors by Abu-Rumeileh et al. have investigated the level of PENK- and PDYN-derived peptides between CJD patients and controls. The results suggest that one of the PENK-derived peptides was significantly decreased in each CJD subtype compared to controls.

The data are clearly presented and the interpretation is reasonable.

There are some major and minor points that the authors may consider:

Major comment

1) The authors analyzed the levels of PENK peptide and PDYN peptide separately from patients and controls (Tables 1,2 and Figure 1). What would be the result if the two levels were combined and analyzed?

Response. We have calculated the mean values of both PDYN and PENK peptides as in our previous paper (Al Shweiki Mov Dis 2020) and included the results of each analysis in Table 1 and supplementary materials. The results obtained largely reflect what we obtained by analyzing each group of peptides separately. We think that the sum of both peptides might be less informative.

2) The authors conducted a study in 66 CJD patients. Among them, the patients with sCJD and fCJD were 63 and 3, respectively. In my opinion, I recommend to analyze excluding three fCJDs, which account for a small part, and then change CJD to sCJD in the title and all contents.

Response. We have excluded the 3 patients from the cohort and updated all analyses and results in the main draft and in the supplementary materials accordingly. We have also changed the title as suggested.  

Minor comments

Line 34, did not associate -> were not associated. Line 44, PRNP -> PRNP. Line 193, PRNP -> PRNP

Response. We corrected the text according to the reviewer’s suggestion.

Line 194, In a sentence “3 carrying a pathogenic PRNP mutation (E200K, V210I, D178N, P102L)”, there are 4 mutation numbers. But why are 3 patients? Patient carrying a PRNP P102L mutation was GSS, not CJD.

Response. Since we have eliminated the genetic cases (see above), the sentence is no longer in the manuscript.

Reviewer 1.

In this study, the authors by Abu-Rumeileh et al. have investigated the level of PENK- and PDYN-derived peptides between CJD patients and controls. The results suggest that one of the PENK-derived peptides was significantly decreased in each CJD subtype compared to controls.

The data are clearly presented and the interpretation is reasonable.

There are some major and minor points that the authors may consider:

Major comment

1) The authors analyzed the levels of PENK peptide and PDYN peptide separately from patients and controls (Tables 1,2 and Figure 1). What would be the result if the two levels were combined and analyzed?

Response. We have calculated the mean values of both PDYN and PENK peptides as in our previous paper (Al Shweiki Mov Dis 2020) and included the results of each analysis in Table 1 and supplementary materials. The results obtained largely reflect what we obtained by analysing each group of peptides separately. We think that the sum of both peptides might be less informative.

2) The authors conducted a study in 66 CJD patients. Among them, the patients with sCJD and fCJD were 63 and 3, respectively. In my opinion, I recommend to analyze excluding three fCJDs, which account for a small part, and then change CJD to sCJD in the title and all contents.

Response. We have excluded the 3 patients from the cohort and updated all analyses and results in the main draft and in the supplementary materials accordingly. We have also changed the title as suggested.  

Minor comments

Line 34, did not associate -> were not associated. Line 44, PRNP -> PRNP. Line 193, PRNP -> PRNP

Response. We corrected the text according to the reviewer’s suggestion.

Line 194, In a sentence “3 carrying a pathogenic PRNP mutation (E200K, V210I, D178N, P102L)”, there are 4 mutation numbers. But why are 3 patients? Patient carrying a PRNP P102L mutation was GSS, not CJD.

Response. Since we have eliminated the genetic cases (see above), the sentence is no longer in the manuscript.

Reviewer 2 Report

The manuscript titled ‘Prodynorphin and proenkephalin in cerebrospinal fluid of Creutzfeldt-Jakob disease’ is interesting and relevant especially in the context of discrimination of subtypes of CJD. There are quite some interesting findings but they are not shown (at least in the supplementary section) and only briefly touched upon. For example, the correlations of PDYN and PENK with established biomarkers or the neuropathological data. The following are the comments that should be specifically addressed to improve the quality of the manuscript.

  1. Introduction: Please provide a brief introduction on the role of PDYN and PENK in the pathology of neurodegenerative diseases. Additionally, a brief introduction on the pathological subtypes of CJD is needed.
  2. Introduction: Line 48-49. Do you mean ‘continuous’ identification? Then please change ‘continue’ to ‘continuous’.
  3. Results: Line 75 and line 104. Please show this AUC data in the supplementary section.
  4. Results: Fig 1. Please improve the figure image quality. Currently, the aspect ratio is not correct which is making it difficult to see the differences among different diagnostic groups.
  5. Results: Fig 2. Please provide some of the representative immunohistochemistry data used for neuropathological analysis for quantification shown in Fig 2.
  6. Results: Lines 124-133. There are some significant correlations with nfL, tau and 14-3-3 that are quite interesting. Please include the plots for all significant correlations in the supplementary section.
  7. Materials and Methods: Please provide the clinical data of the patients (sCJD MM(V)1 (n = 31), VV2 (n = 15) and MV2K (n = 13) brains) used for the neuropathological analysis in the supplementary section. Were these patients same (or subset) of patients whose CSF samples were analyzed?
  8. Discussion: A speculation or hypothesis on why the peptide levels may differ among different subtypes of CJD (in the context of brain pathology and the rate of disease progression) should be included.

Author Response

Reviewer 2.

The manuscript titled ‘Prodynorphin and proenkephalin in cerebrospinal fluid of Creutzfeldt-Jakob disease’ is interesting and relevant especially in the context of discrimination of subtypes of CJD. There are quite some interesting findings but they are not shown (at least in the supplementary section) and only briefly touched upon. For example, the correlations of PDYN and PENK with established biomarkers or the neuropathological data. The following are the comments that should be specifically addressed to improve the quality of the manuscript.

  1. Introduction: Please provide a brief introduction on the role of PDYN and PENK in the pathology of neurodegenerative diseases. Additionally, a brief introduction on the pathological subtypes of CJD is needed.

Response. We implemented the introduction addressing the issues indicated by the reviewer (lines 47-49 and 61-68).

  1. Introduction: Line 48-49. Do you mean ‘continuous’ identification? Then please change ‘continue’ to ‘continuous’.

Response. We changed the text accordingly (line 52).

  1. Results: Line 75 and line 104. Please show this AUC data in the supplementary section.

Response. We included the AUC data for all comparisons in the supplementary section.

  1. Results: Fig 1. Please improve the figure image quality. Currently, the aspect ratio is not correct which is making it difficult to see the differences among different diagnostic groups.

Response. We tried to improve the figure quality by standardizing/levelling as much as possible the scales between boxplots.

  1. Results: Fig 2. Please provide some of the representative immunohistochemistry data used for neuropathological analysis for quantification shown in Fig 2.

Response. We used H&E stained slides for the neuropathological analysis. We have provided some representative sections used for this analysis in the supplementary material.  

  1. Results: Lines 124-133. There are some significant correlations with nfL, tau and 14-3-3 that are quite interesting. Please include the plots for all significant correlations in the supplementary section.

Response. We added the plots showing significant correlations with the neurodegenerative biomarkers in the supplementary materials.

  1. Materials and Methods: Please provide the clinical data of the patients (sCJD MM(V)1 (n = 31), VV2 (n = 15) and MV2K (n = 13) brains) used for the neuropathological analysis in the supplementary section. Were these patients same (or subset) of patients whose CSF samples were analyzed?

Response. For neuropathological data, we only used patients who had their CSF analyzed for PDYN and PENK levels. They comprised 56 of the 63 sCJD patients included in the study. Given there is only a slight discrepancy between the two groups, we have provided the clinical data for the whole cohort, including the 6 patients with “highly” probable sCJD (all positive by prion CSF RT-QuIC) in the supplementary material (Table S5).   

Discussion: A speculation or hypothesis on why the peptide levels may differ among different subtypes of CJD (in the context of brain pathology and the rate of disease progression) should be included.

Response. We have already discussed the possible influence of the severity of neurodegeneration in the basal ganglia and cerebral cortex on PDYN and PENK levels. In contrast, the rate of disease progression does not seem to play a significant role given the lack of association between peptide levels and survival in sCJD. We have addressed this issue in the discussion (lines 188-192).

Round 2

Reviewer 1 Report

The revised manuscript was markedly improved and satisfactorily modified for reviewer comment. Thus, I am recommending acceptance for publication in the IJMS.

Reviewer 2 Report

The authors have addressed all the previous comments.